# Overview of Reptile Diversity from Bobaomby Complex, Northern Tip of Madagascar

**DOI:** 10.3390/ani13213396

**Published:** 2023-11-01

**Authors:** Raphali R. Andriantsimanarilafy, Alain J. V. Rakotondrina, Josué A. Rakotoarisoa, Ranaivoson T. Nasaina, Jeanneney Rabearivony, Achille P. Raselimanana

**Affiliations:** 1Association Groupe d’Etudes et de Recherche sur les Primates de Madagascar, Lot II M 78 Bis Antsakaviro, BP 779, Antananarivo 101, Madagascar; 2Madagasikara Voakajy, BP 5181, Antananarivo 101, Madagascar; arraphali@voakajy.mg; 3Fondation Tany Meva, Lot I A I 1bis Ambatobe, BP 4300, Antananarivo 103, Madagascar; josue.rakotoarisoa@gmail.com; 4Mention Zoologie et Biodiversité Animale, Facultés des Sciences, Université d’Antananarivo, BP 906, Antananarivo 101, Madagascar; tambyranaivoson@gmail.com; 5Facultés des Sciences, Université d’Antsiranana, BPO, Antsiranana 201, Madagascar; r.jeanneney@gmail.com; 6Association Vahatra, BP 3972, Antananarivo 101, Madagascar; raselimananaachille@gmail.com

**Keywords:** reptile survey, deciduous forest, micro-endemic, protected area, northern Madagascar

## Abstract

**Simple Summary:**

Madagascar is home to diverse ecosystems with many endemic reptiles, many of which are threatened by the increasing loss of their habitat. Unfortunately, while some areas have been relatively well studied, there is a paucity of data in most of them. This knowledge gap hinders efforts to collect scientific information in order to conserve the remaining habitat. We conducted surveys in the Bobaomby Complex, in the northern tip of Madagascar to assess its potential for the creation of a new protected area. We found 42 reptile species of which 39 are endemic and the discovery of many threatened species with restricted range distributions. The findings fill the knowledge gaps on the herpetofauna of the Bobaomby Complex. We recommend the inclusion of the Bobaomby Complex into the network of protected areas of Madagascar.

**Abstract:**

Many studies on reptiles have been conducted across Madagascar but some areas are poorly known in terms of the diversity of reptiles such as the Bobaomby Complex in the northern tip of Madagascar. In February and March 2018, we conducted a biodiversity survey within five sites. This biological survey is to collect scientific information for helping new protected creations. Three main methods were used including pitfall trap, visual and acoustic searching along the transect and refuge examination. In total, we recorded 42 species including 5 chameleons, 8 skinks, 11 geckos, 16 snakes and 2 blinds snake species. All recorded species are endemic to Madagascar except *Hemidactylus frenatus*, *Ebenavia inunguis* and *Phelsuma abbotti*. Rare species known only from a few specimens have been recorded in the Bobaomby Complex: *Heteroliodon fohy*, *Pseudoxyrhopus ambreensis*, and *Madascincus arenicola*. Thirteen species are classified as threatened on the IUCN Red List, of which three are Critically Endangered: *Paracontias minimus*, *Madascincus arenicola*, and *Paroedura lohatsara*; three are Endangered: *Heteroliodon fohy*, *Lycodryas inopinae*, and *Phisalixella variabilis*; and seven are Vulnerable: *Brookesia ebenaui*, *Furcifer petteri*, *Blaesodactylus boivini*, *Uroplatus ebenaui*, *Uroplatus henkeli*, *Liophidium therezieni* and *Flexiseps ardouini*. Our results reveal the importance of the Bobaomby Complex for conserving reptile diversity and highlight the need to protect it.

## 1. Introduction

Madagascar is remarkable for its diversified ecosystems and a high rate of species endemism [1]. The island is home to 436 native non-marine reptile species [2] and with more than 99% of endemicity [3]. However, many species are threatened by habitat loss and fragmentation caused by expanding agriculture as well as logging and wood harvesting [4]. Based on the IUCN Red List assessment, at least 29.9% of Malagasy reptile species are classified as threatened [5].

Protected areas (PAs) are crucial worldwide to ensure the conservation of biodiversity in the face of loss and fragmentation of ecosystems [6]. According to the Durban Vision in 2003, the Malagasy government committed to tripling the surface area of PAs in Madagascar in order to guarantee the long-term conservation of its rich and unique biodiversity. Since this declaration, many new PAs have been created, mainly encompassing humid rainforests of the eastern part of the island. However, other ecosystems are far less well-represented in the Malagasy PA system. For example, many dry forests in Madagascar remain unprotected although they are among the most threatened ecosystems on the island [7]. Moreover, Malagasy dry forests are known for their rich herpetofauna communities, but many sites have not received extensive attention from researchers and their herpetofauna diversity is poorly known [8]. Protected areas have been subject to more survey efforts than unprotected areas [9]. Further exploration of these understudied sites is crucial to provide valuable information on biodiversity to inform future conservation efforts. This study aims to provide scientific information on a reptile community to establish a management and conservation plan for sites without current protection. Indeed, the biological exploration of forests outside PAs is essential [10] to assess its importance for conservation, as in the case of the Bobaomby Complex in the extreme north of Madagascar. Previous research here in the form of a biological survey of the Ampombofofo and Anjiabe forest sites has resulted in the finding of 6 endemic amphibians and 37 endemic reptiles [11]. These findings prompted the designation of these sites as Key Biodiversity Areas due to the presence of three Critically Endangered, two Endangered and one Vulnerable species [10]. However, some sites within the Bobaomby Complex such as Beantely, Antsisikala and Ambanililabe are poorly studied. Here, we report findings from surveys we carried out to assess the importance of the Bobaomby Complex in terms of the diversity of reptiles to support the proposal of a new protected area.

## 2. Materials and Methods

### 2.1. Study Location

The Bobaomby Complex is located in the extreme north of Madagascar (Figure 1), District Antsiranana II, Diana Region. The reptile survey was carried out in the remaining forest of the Bobaomby Complex from February to March 2018, a hot and rainy season coinciding with optimal biological activities for most species and favorable to their census [12,13]. Five sites, Beantely, Antsisikala, Ambanililabe, Anjiabe and Ampombofofo (Figure 1), were inspected. Each site was visited for five successive days.

The study area belongs to the western dry deciduous forest ecoregion [14] and includes the northern part of the island. The vegetation has a canopy of 10 to 15 m, sometimes reaching 20 m. Rocky or karst formations backed by dense deciduous vegetation and tree savannahs characterize the landscape in the Bobaomby Complex (Figure 2). The vegetation and habitat characteristics of the study sites were assessed from field observations (Table 1).

### 2.2. Sampling Techniques

We employed (i) pitfall trapping with drift fences, (ii) visual searching, and (iii) refuge examination [15] methods to sample animals.

#### 2.2.1. Pitfall Traps

This method is intended for the capture of burrowing and terrestrial reptiles [11]. The pitfall traps were buckets (275 mm deep, 290 mm top internal diameter, 220 mm bottom internal diameter) with the handles removed and small holes (2 mm diameter) punched in the bottom to allow water drainage. Buckets were sunk into the ground below a drift fence made from plastic sheeting (0.5 m high) stapled in a vertical position to thin wooden stakes, with the fence bottom buried 50 mm deep into the ground using soil and leaf litter (Figure 3). The drift fence (100 m in length) was positioned to run across the middle of each pitfall trap. A pitfall trap was positioned at both ends of the drift fence, with the other nine traps at 10 m intervals. At each site, three lines of tarps were used, except in Anjiabe, where we deployed four to increase our effort proportionally with the size of the forest fragment. Lines were placed in each of the following forest types: ridge (along the crest of a ridge), slope (on a gradient, intermediate between ridge top and valley bottom) and valley (within 20 m of a stream in a valley bottom). At each site, sampling was conducted for five days. The traplines were checked every morning (06:00 a.m.) and late afternoon (06:00 p.m.).

#### 2.2.2. Visual Searching

Animals were sampled throughout transects of 150 m length which are composed of three parallel lines of 50 m, separated 20 m apart and side by side [16]. Two transects are separated at least by a distance of 200 m. Each transect is visited once during the day and once during the night for five consecutive days. Diurnal searches were conducted between 7:30 a.m. and 12:00 p.m., periods when reptiles are more active [16]. Night searches were carried out between 07:00 to 10:00 p.m. using headlamps (Petzl MYO RXP) to spot animals. The survey team was composed of three experienced herpetologists.

#### 2.2.3. Refuge Examination

Several species retreat to refuge when inactive [17]. Refuge examination was performed along a transect line during the day. Microhabitats likely to be a refuge were examined: under and in fallen logs and rotten tree stumps, removal bark, rocks crevasses, in leaf litter, root mats and soil, among dead wood, in leaf axils of *Pandanus* screw palms and *Ravenala* traveler’s palm.

For each encountered reptile, the following variables were recorded: time of sighting, GPS coordinates, substrate type and species names. At least two individuals of each species were pictured for documentation of the natural coloration of the animal in its habitat and to serve as later identification [18]. Species identification was based on morphology and on an expert-based assessment according to the key descriptions by Glaw and Vences [18] and the nomenclature based on current taxonomy. Two specimens for each species that are difficult to identify were collected and preserved in 70% ethanol [19].

### 2.3. Statistical Analysis

To have global insight into the populations within the community, raw counts for each taxon were given at each site. The species recorded from the area are classified using a system similar to that used by Wilson and McCranie [20] and can be summarized as follows: abundant (large numbers encountered on a regular basis); common (encountered on a regular basis); infrequent (unpredictable, few individuals seen); or rare (rarely seen). These classifications are based on data collected using active searching, refuge examination and pitfall traps.

## 3. Results

### 3.1. Species Richness and Composition

A total of 42 reptile species were recorded during a total effort of 25 days of active search (Table 2). Anjiabe and Ampombofofo had the highest number of encountered species with 31 and 29 species, respectively, followed by Beantely with 26 species. Ambanililabe and Antsisikala had the lowest number with 18 and 17 species, respectively (Figure 4).

None of the curves reached the plateau before the end of the investigation period, meaning that we may have not observed some species present at the study sites (Figure 4). Therefore, the Bobaomby Complex could contain more species of reptiles than we encountered during this study. The effort undertaken was apparently not sufficient to identify all the species in the study sites.

The reptile community in this area was dominated by lizards with 21 species (50%), 16 species of snakes (38.09%), and 5 species of chameleon species representing 11.9% of our total species (Table 2).

Most species are inventoried with multiple survey methods. However, some species were only recorded with the refuge examination. These species include *Ithycyphus miniatus*, *Ebenavia inunguis*, *Paracontias minimus*, *Paracontias* sp. aff. *rothschildi*, and *Paracontias sp.* Four species were collected only by pitfall trap: *Flexiseps ardouini*, *Heteroliodon fohy*, *Madascincus arenicola*, and *Liophidium therezieni*. The combination of these three techniques was useful for investigating the reptile community across different habitat types and species behaviors.

### 3.2. Distribution and Conservation Status

Among the encountered reptile species, 13 were classified as threatened according to the IUCN Red List (Table 2), including seven listed as Vulnerable: *Brookesia ebenaui*, *Furcifer petteri*, *Blaesodactylus boivini*, *Uroplatus ebenaui*, *Uroplatus henkeli*, *Flexiseps ardouini*, and *Liophidium therezieni*; three are Endangered: *Heteroliodon fohy*, *Lycodryas inopinae*, and *Phisalixella variabilis*; and three are Critically Endangered: *Paracontias minimus*, *Madascincus arenicola* and *Paroedura lohatsara*. One species recorded is listed as Near Threatened: *Pseudoxyrhopus ambreensis*. These findings indicate that the Bobaomby Complex represents a potential refuge that can provide protection for these threatened species. Seven species with microhabitats in northern Madagascar were inventoried in the Bobaomby Complex: *Heteroliodon fohy*, *Lycodryas inopinae*, *Liophidium therezieni*, *Madascincus arenicola*, *Paracontias minimus*, *Paroedura lohatsara* and *Pseudoxyrhopus ambreensis*.

### 3.3. Species Raw Counts

The raw counts of reptile species varied among sites (Table 2). Overall, reptiles in the Bobaomby Complex are rare species represented by less than five individuals. One species (*Geckolepis maculata*) was found to be common or abundant at all of the study sites. Some species were found to be rare for some sites and common for others (Table 2): *Blaesodactylus Boivini*, *Brookesia stumpffi* and *Paroedura stumpffi*.

## 4. Discussion

### 4.1. Species Richness and Composition

Each reptile group is well represented at Bobaomby Complex. With its 42 species of reptiles, the Bobaomby Complex hosts more species compared to other sites in the northern of Madagascar. Orangea forest, located 10 km east of the Bobaomby Complex, is home to 22 species of reptiles [11], of which 81.8% are found in the Bobaomby Complex. The forest fragments between the national parks of Montagne d’Ambre and Ankarana host only 34 species of reptiles [7], and the Analamerana Special Reserve contains 32 species [12]. However, the Bobaomby Complex contains lower reptile richness than Montagne des Français hosting 52 reptile species [21]. We are conscious that differences in sampling techniques and effort may influence the number of species recorded across these studies.

In our study, the shape of the cumulative curves of the species at each site clearly shows that additional species could be added and the plateau is still far from being reached. Ramanamanjato et al. [22] reported that the sampling effort for herpetofauna surveys was important for accurate species inventories and suggested increasing the sampling period to nine days to ensure that all species at a given site are surveyed. Other species such as *Hemidactylus mercatorius*, *Zonosaurus boettgeri*, *Trachylepis tavaratra*, *Langaha madagascariensis*, *Xenotyphlops grandidieri* and *Pelusios castanoides* have been recorded in the Ampombofofo site [11], but were not recorded during the present study. As per inventory, no new species of reptile was encountered during our investigation. In fact, some of the habitat was not visited during our investigation due to logistical challenges. In addition, the weather conditions during this study, which included the passage of two cyclones, likely influenced our sampling.

The results of this investigation constitute a first overview of the reptile community for three of our study sites (Beantely, Antsisikala and Ambanililabe) and serve as a database that will fill the knowledge gaps on the reptiles of the northern part of Madagascar.

### 4.2. Ecological Characteristics of the Reptile Community

The reptile community of the Bobaomby Complex is composed of chameleons, lizards, ophidians, skinks and blind snakes. This indicates that there is a diversification of the habitats existing in this ecosystem. The reptile community is represented by arboreal (52.4%), terrestrial (33.3%) and burrowing (9.5%) species. Differences in the occurrence of different taxa between sites suggest a close relationship between the diversity of the ecological environment and the sensitivity of some groups to habitat disturbance. The majority of burrowing species were encountered in Anjiabe and Ampombofofo. These sites are characterized by a particularity of the substrate formed by sandy and soft soil with thick litter which constitutes a unique and suitable habitat for burrowing species, especially skinks [23]. Some species were observed only in more specific habitats. This is the case of arboreal and nocturnal snakes *Phisalixella variabilis*, *P. arctifaciata*, *Lycodryas inopinae* and *L. granuliceps* which were only encountered in forests with rock crevices or karstic formations [18]. Other species have the ability to adapt to different environments and biotopes such as *Blaesodactylus boivini* and *Phelsuma grandis* which have been observed in the forest and also in some very disturbed habitats.

### 4.3. Endemicity and Species Conservation Status

The majority of reptile species recorded in the Bobaomby Complex (92.8%) are all endemic to Madagascar except *Hemidactylus frenatus*, *Ebenavia inunguis*, *Phelsuma abbotti* [17] and *Indotyphlops braminus*. Six inventoried species *Flexiseps ardouini*, *Lycodryas inopinae*, *Heteroliodon fohy*, *Uroplatus ebenaui*, *Phisalixella variabilis* and *Furcifer petteri* are regionally endemic to northern Madagascar. Also, some species with restricted range and only known from a few localities are recorded in the Bobaomby Complex: *Paracontias minimus*, which was previously collected from Orangea [11], *Paroedura lohatsara* and *Pseudoxyrhopus ambreensis* in the Montagne des Français [22], and *Madascincus arenicola* in Baie des Dunes, Ramena [24]. The tree snake *Lycodrias inopinae*, recorded in Anjiabe and Ampombofofo, was previously known in the Montagne des Français [21].

### 4.4. Extension of Distribution Area

The Bobaomby Complex represents an extension of the distribution area for *Liophidium therezieni*, *Brookesia ebenaui*, *Heteroliodon fohy*, *Paroedura lohatsara* and *Pseudoxyrhopus ambreensis*. These species have been reported among the four localities including the Montagne des Français, Anatelo, and Forêt d’Orangea [9,11,21] and the identification of *B. ebenaui* at Beantely forest represented an extension of the distribution area for this species [25]. *B. ebenaui* was previously known in the Amber Forest [15], in the Montagne des Français [21] and in Ankarana [7].

### 4.5. Conservation Value

Given that herpetofauna constitutes an element in the choices of process identifying priority sites in terms of conservation [26,27], Beantely, Anjiabe and Ampombofofo forests are the potential sites for the main conservation zone of the future protected area because they are home to numerous threatened and endemic reptiles. They also constitute a remaining bloc of forest in the northern part of Madagascar and represent a new recorded range extension for some species. In total, thirteen species of reptiles (30.9%) from the Bobaomby Complex are on the IUCN Red List, of which three species are classified as Critically Endangered, three as Endangered and seven as Vulnerable. In addition, the records of four species indicate a range extension more than 20 km north of their previously known distribution [15,21]. The Bobaomby Complex may play an essential role in maintaining the population of reptiles in dry deciduous forests [24]. Reptiles are also among the groups most vulnerable to extinction due to their environmental requirements and their dependence on forest ecosystems [28]. Therefore, the creation of a new protected area of this site is strongly recommended with the inclusion of Beantely, Anjiabe and Ampombofofo as a core area according to its criteria for conservation prioritization [28,29]; as mentioned by Andreone, due to its natural richness, northern Madagascar is already a key area of conservation in the region [30].

## 5. Conclusions

The investigation we carried out in the Bobaomby Complex provides new insights into the reptilian diversity of the area. This investigation was the first to catalog the reptile communities of Beantely, Antsisikala and Ambanililabe. Among the 42 species recorded in the Bobaomby Complex, 13 species are threatened with extinction and 11 are endemic to the area. The Bobaomby Complex is rich in reptiles and plays an important role in the conservation of biodiversity representative of the northern part of Madagascar. The reptilian community of the Bobaomby Complex is characterized by the abundance of arboreal species. The geographical location, the heterogeneity of natural habitats, and the presence of micro-endemic and threatened species highlights the ecological importance of this site. The integration of the Bobaomby Complex into the network of protected areas of Madagascar will contribute to the maintenance of natural ecosystems and the conservation of reptilian diversity. We recommend the inclusion of Beantely, Anjiabe and Ampombofofo as the core area of the new protected area. The results of this investigation will serve as a database to support the establishment of a future protected area and fill in a knowledge gap on reptiles in the region. However, molecular analysis is recommended to confirm the taxonomy of some species which might be a new species in this region.

### Photos of Reptiles from Bobaomby Complex, Captured during Fieldwork

A photographic repertoire of some species recorded is provided (Figure 5, Figure 6 and Figure 7) to illustrates the repitles species within Bobaomby Complex during this investigation.

## Figures and Tables

**Figure 1 animals-13-03396-f001:**
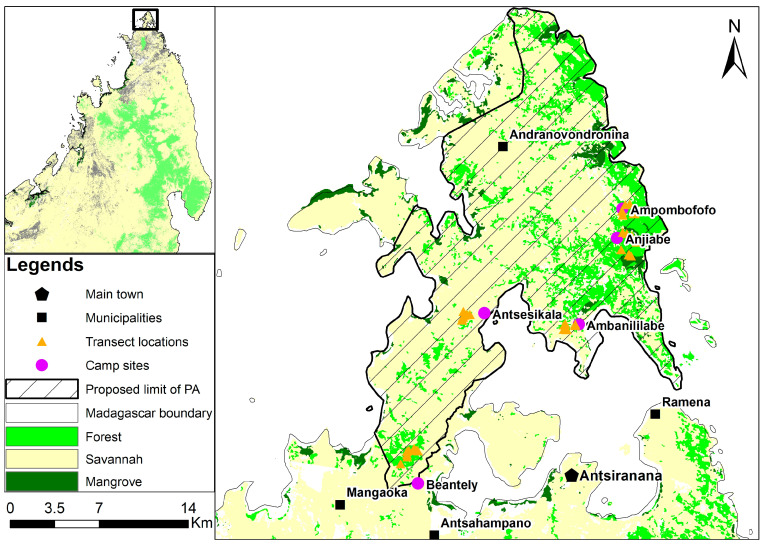
Location of the study sites.

**Figure 2 animals-13-03396-f002:**
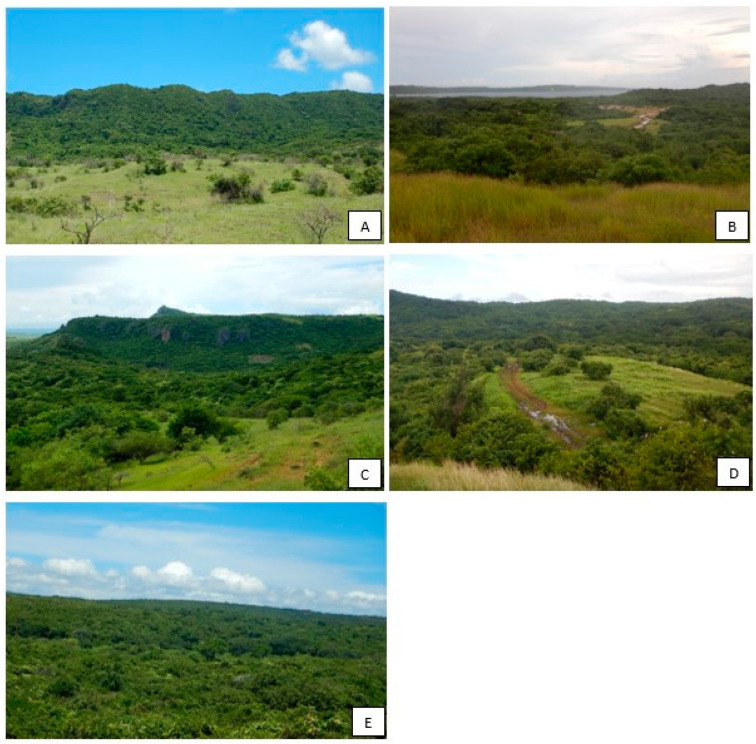
Different types of habitat within the five sites. (**A**) Beantely, a primary forest dominated by savannah; (**B**) Ambanililabe, a degraded forest surrounded by a savannah and mangrove forest; (**C**): Antsisikala, a disturbed forest surrounded by a large savannah; (**D**): Anjiabe, dominance of intact forest; (**E**): Ampombofofo, primary and relatively intact forest.

**Figure 3 animals-13-03396-f003:**
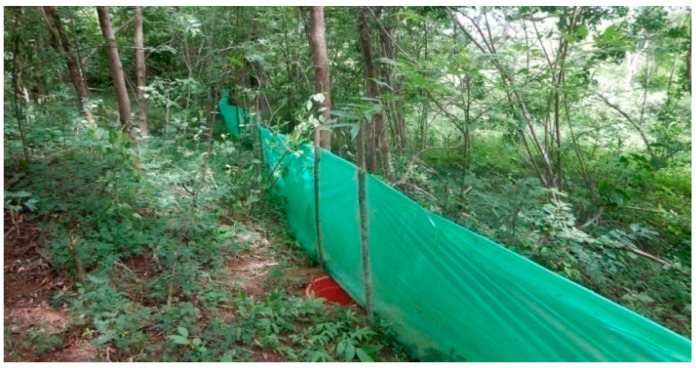
Pitt fall trap line used to capture reptiles.

**Figure 4 animals-13-03396-f004:**
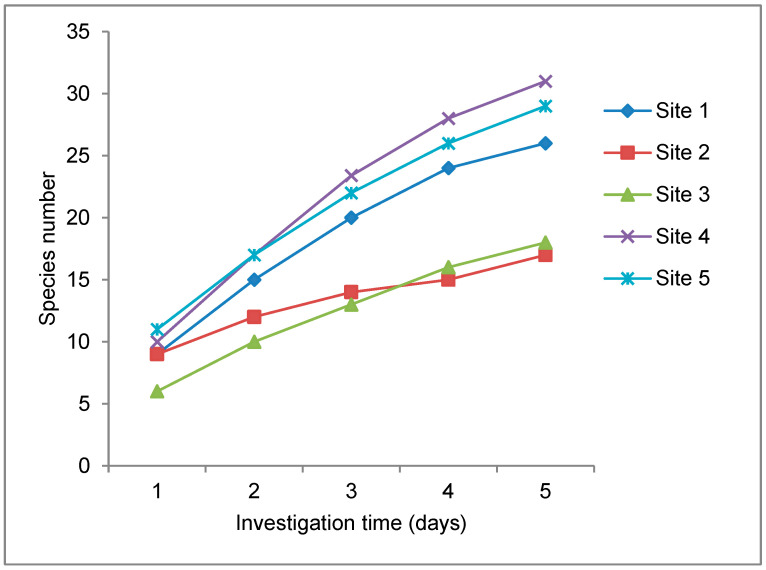
Accumulation curves of encountered reptile species in Bobaomby Complex.

**Figure 5 animals-13-03396-f005:**
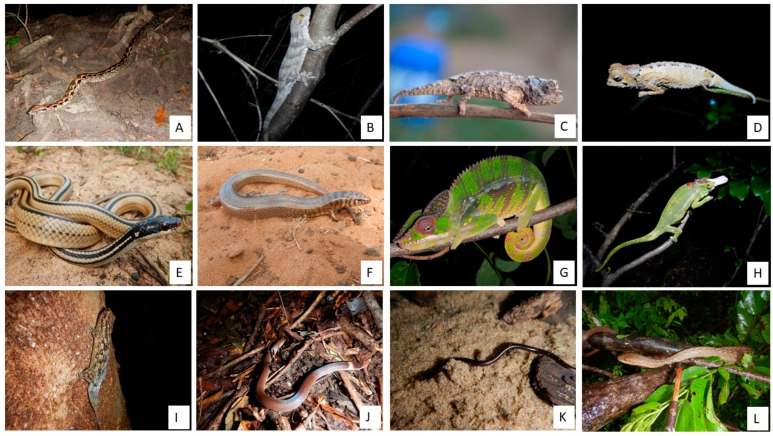
Reptile species of Bobaomby Complex: (**A**) *Acrantophis madagascariensis*, (**B**) *Blaesodactylus boivini*, (**C**) *Brookesia ebenaui*, (**D**) *Brookesia stumpffi*, (**E**) *Dromicodryas quadrilineatus*, (**F**) *Flexiseps ardouini*, (**G**) *Furcifer pardalis*, (**H**) *Furcifer petteri*, (**I**) *Geckolepis maculata*, (**J**) *Heteroliodon fohy*, (**K**) *Indotyphlops braminus*, (**L**) *Ithycyphus miniatus*.

**Figure 6 animals-13-03396-f006:**
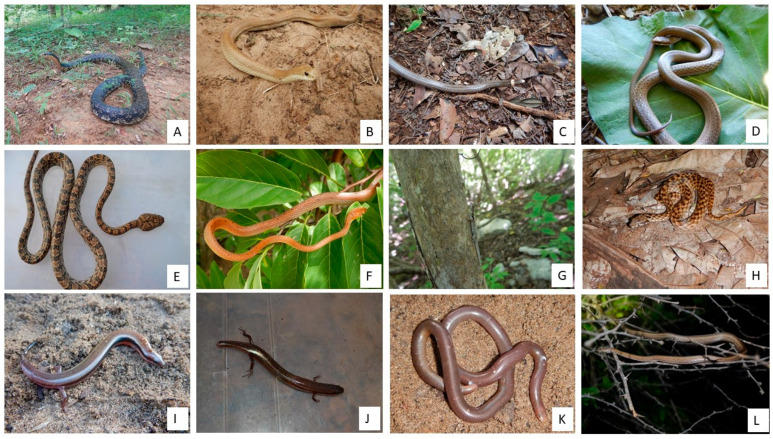
Reptile species of Bobaomby Complex: (**A**) *Leioheterodon madagascariensis*, (**B**) *Leioheterodon modestus*, (**C**) *Liophidium therezieni*, (**D**) *Liophidium torquatum*, (**E**) *Lycodryas granuliceps*, (**F**) *Lycodryas inopinae*, (**G**) *Lygodactylus heterurus*, (**H**) *Madagascarophis colubrinus*, (**I**) *Madascincus arenicola*, (**J**) *Madascincus intermedius*, (**K**) *Madatyphlops mucronatus*, (**L**) *Mimophis occultus*.

**Figure 7 animals-13-03396-f007:**
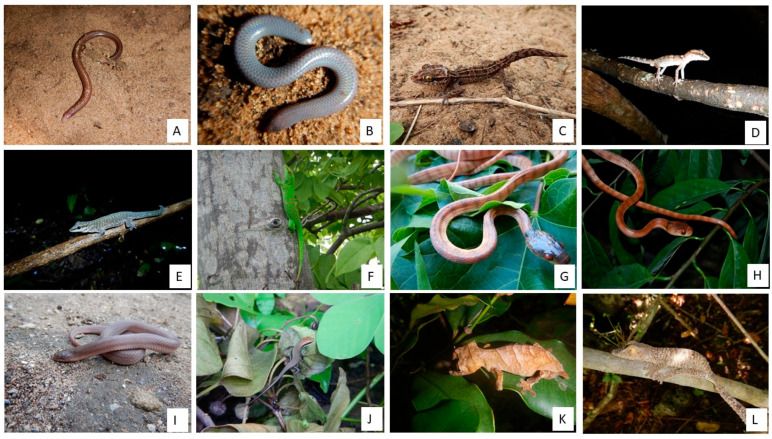
Reptile species of Bobaomby Complex: (**A**) *Paracontias minimus*, (**B**) *Paracontias* sp., (**C**) *Paroedura lohatsara*, (**D**) *Paroedura stumpffi*, (**E**) *Phelsuma abbotti*, (**F**) *Phelsuma grandis*, (**G**) *Phisalixella arctifasciata*, (**H**) *Phisalixella variabilis*, (**I**) *Pseudoxyrhopus ambreensis*, (**J**) *Trachylepis gravenhorstii*, (**K**) *Uroplatus ebenaui*, (**L**) *Uroplatus henkeli*.

**Table 1 animals-13-03396-t001:** Specific characteristics of the vegetation in the study sites.

Sites	Geographic Coordinates	Altitude (m)	Vegetation Characteristics
Beantely (Site 1)	12°16′33.2″ S 49°10′05.7″ E	119–200	Disturbed forest, closed canopy (10–20 m height), diffuse undergrowth, thick litter, clay and rocky soils, temporary streams
Antsisikala (Site 2)	12°10′31.8″ S 49°12′56.5″ E	53–150	Disturbed forest, open canopy (5–8 m height), light undergrowth, thin or even absent leaf litter, clay and rocky soils, temporary streams
Ambanililabe (Site 3)	12°11′21.4″ S 49°17′33.7″ E	11–50	Disturbed forest located near coastal zone, open canopy (5–8 m high), light undergrowth, thin or even absent leaf litter, clay and rocky soils, temporary streams
Anjiabe (Site 4)	12°07′02.6″ S 49°20′06.8″ E	11–91	Intact forest, semi-open canopy (5–15 m high), abundant undergrowth, very thick litter, clay or sandy soils, temporary stream
Ampombofofo (Site 5)	12°05′38.76″ S 49°20′23.89″ E	24–70	Relatively intact forest, closed canopy (5–15 m high), well-stocked undergrowth, very thick litter, sandy soils, permanent watercourse, presence of marshes and ponds

**Table 2 animals-13-03396-t002:** List and raw counts of reptiles in the Bobaomby Complex. Captured by: VS = visual survey, PT = pitfall trapping, RE = refuge examination. IUCN Status: DD = data deficient, LC = least concern, NE = not evaluated, NT = Near Threatened, VU = Vulnerable, EN = Endangered, CR = Critically Endangered, Endemicity: E = endemic to Madagascar, Er = endemic regional, N = not endemic. Ecological distribution: TE = terrestrial, AB = arboreal, BR = burrowing.

Taxa	Captured by	IUCN Status	Endemicity	Ecological Distribution	Species Raw Counts					Distribution	New Distribution	Voucher Code
					Site 1	Site 2	Site 3	Site 4	Site 5			
SANZINIIDAE												
*Acrantophis madagascariensis*	VS	LC	E	TE	1	1		1	1	North		
*Sanzinia volontany*	VS	NE	E	AB, TE					1	West, north, south		
CHAMAELEONIDAE												
*Brookesia ebenaui*	VS	VU	Er	AB	3					North and northwest	Bobaomby Complex	UADBA-R-71739
*Brookesia stumpffi*	VS	LC	E	AB	20			8	21	North and northwest		UADBA-R-71736
*Furcifer oustaleti*	VS	LC	E	AB	2	1	1	1	2	All of Island, except central south		
*Furcifer pardalis*	VS	LC	E	AB	4	6	1	1	5	Northwest, north, east		
*Furcifer petteri*	VS	VU	Er	AB	2			18	4	North and northwest		UADBA-R-71740
LAMPROPHIIDAE												
*Dromicodryas quadrilineatus*	VS	LC	E	TE	1		1	1	1	North and east		
*Heteroliodon fohy*	PT	EN	E	TE	2			3	1	North		UADBA-R-71747
*Ithycyphus miniatus*	RE	LC	E	AB				1		West and north		
*Leioheterodon madagascariensis*	VS	LC	E	TE	1	1	1	1	1	Much of Island		
*Leioheterodon modestus*	PT, VS	LC	E	TE			1	1		North, west, south		
*Liophidium therezieni*	VS	VU	E	TE				1		North	Bobaomby Complex	UADBA-R-71745
*Liophidium torquatum*	VS	LC	E	TE		1	2	2	2	Much of Island, except south		UADBA-R-71748
*Lycodryas granuliceps*	VS	LC	E	AB	1		2	1	1	North		
*Lycodryas inopinae*	VS	EN	ER	AB				1	1	North		UADBA-R-71744
*Madagascarophis colubrinus*	VS	LC	E	TE	1	4	1		1	Much of Island		
*Phisalixella arctifasciata*	VS	LC	E	AB					R	East		
*Phisalixella variabilis*	VS	EN	E	AB	1				1	North and northwest		UADBA-R-71741
*Pseudoxyrhopus ambreensis*	VS	NT	Er	TE			2		1	North	Bobaomby Complex	UADBA-R-71749
PSAMMOPHIIDAE												
*Mimophis occultus*	VS	-	E	TE	1	1	1			North, northwest, west		
GEKKONIDAE												
*Blaesodactylus boivini*	VS	VU	Er	AB	6	32	3	4	2	North		UADBA-R-71731
*Ebenavia inunguis*	RE, VS	LC	N	AB				1		Northwest		
*Geckolepis maculata*	RE, VS	LC	E	AB	13	17	28	11	6	Northern and south		UADBA-R-71728
*Hemidactylus frenatus*	VS	LC	N	AB	2		1		1	North, west, south		UADBA-R-71727
*Lygodactylus heterurus*	VS	LC	Er	AB	1	2				North and northwest		UADBA-R-71724
*Paroedura lohatsara*	PT, VS	CR	Er					2	1	North	Bobaomby Complex	UADBA-R-71725
*Paroedura stumpffi*	VS	LC	E	TE, AB	2	14	3	1	1	North and northwest		UADBA-R-71726
*Phelsuma abbotti*	VS	LC	N	AB	1	12		3	4	North, west, Seychelles		UADBA-R-71721
*Phelsuma grandis*	VS	LC	E	AB	3	4	3	5	5	North		UADBA-R-71722
*Uroplatus ebenaui*	VS	VU	E	AB	2					North and northwest		UADBA-R-71718
*Uroplatus henkeli*	VS	VU	E	AB	1		1	1	3	Northwest		UADBA-R-71720
SCINCIDAE												
*Flexiseps ardouini*	PT	VU	E	TE				1	3	North		UADBA-R-71732
*Madascincus polleni*	PT	LC	E	TE	1	1	1	1		West		UADBA-R-71705
*Madascincus arenicola*	PT	CR	Er	TE, BR				13	10	North		UADBA-R-71704
*Paracontias minimus*	RE	CR	Er	BR				8	4	North		UADBA-R-71756
*Paracontias* sp.	RE		Er	BR				1		North		UADBA-R-71753
*Paracontias* sp. aff. *rothschildi*	RE		Er	BR				1		North		UADBA-R-71751
*Trachylepis elegans*	VS	LC	E	TE		2	8		3	Much of Island		UADBA-R-71707
*Trachylepis gravenhorstii*	VS	LC	E	TE	2	12		3	2	Much of Island		
TYPHLOPIDAE												
*Madatyphlops mucronatus*	VS	DD	E	BR				1		Northwest		UADBA-R-71703
*Indotyphlops braminus*	VS	-	E	BR	1	2		2		Much of Island		UADBA-R-71719
Total reptiles: 42					26	17	18	31	29			

## Data Availability

https://bobaombyreptile.org/01.2018/001 (accessed on 9 May 2023).

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
