# Peer review of "Overview of Reptile Diversity from Bobaomby Complex, Northern Tip of Madagascar"

_animals, 2023, doi:10.3390/ani13213396_

Round 1

Reviewer 1 Report

Comments and Suggestions for Authors

In this study, the authors inventoried the reptile community of the area of Complexe Bobaomby, in northern Madagascar. Upon intensive field samplings, they identified the detected species based on morphology. By doing this, they provide a new species list for a wide area previously only partially inventoried.

The study presents new relevant information on the knowledge of the distribution of Malagasy reptiles. I found the project design and applied methodology appropriate for the study aims, and the manuscript is well-structured. However, I recommended “Reconsider after major revision” because of a few issues, consisting of a few relevant and mostly minor questions related to the manuscript presentation.

Among the relevant issues, I have found the lack of animal pictures a major oversight. I expected to see photographic plates reporting all sampled species, which would help corroborate species identification. In case the authors did not take pictures, this would also be a major methodological flaw, especially considering they identified species based on morphology. I also expected pictures of the sampling sites to understand the habitat types. Another important aspect is the use of quite old references. For example, the authors did not consider recent major publications on Malagasy herpetofauna (and not only) like “The New Natural History of Madagascar” and “The Terrestrial Protected Areas of Madagascar: Their History, Description, and Biota”, which are appropriate to contextualise species lists and species inventories. They should generally use more recent references throughout the manuscript. Finally, the manuscript requires a major English revision.

Besides these major aspects, you can find below a list of minor questions I highlighted in the text:

 4 – Missing first author’s first name

20- Delete “all” and substitute” discovered” with “discovery”

23- Instead of distribution better using “species cataloguing”

41- Endemism of which groups? Animals, plants? Please, specify

42- 440 is the number of reptile species, but herpetofauna refers also to amphibians. Please, correct

42- 440 refers to native species. Please, add “native”

42- Referring to only reptiles it should be 99% of endemic native non-marine species. Please, correct

43- It is not only habitat loss in general but also forest fragmentation. I suggest adding something about forest fragmentation and further developing this point describing threats to Malagasy reptiles.

44- “as threatened with extinction” Do you refer to the three threatened IUCN categories VU, EN, CR? Please, specify

45- This sentence is missing any reference. I also think you should further develop this concept, which is important to the study.

50-52 Missing reference.

57- The sentence is incomplete. Please add "on the biological community inhabiting the area" at the end of the sentence or something similar.

76- There is no legend of Figure 1

87 - 2.2. Sampling techniques section. Please, specify what you did when you found the animals (e.g., taking pictures, coordinates of the point). Have you released the animals upon manipulation and identification in loco? Please, specify. Have you collected any voucher specimens from those species that are difficult to identify and/or have you collected tissue samples for possible future DNA Barcoding and genetic analyses? In case you did, please specify.

88- You said something similar a few lines before, which is redundant. Keep it either here or there

90- I am not sure that “acoustic searching” is the right term as you did not survey amphibians. You can keep only “visual”

106- Same as for line 90

108- “parallel lines”. If the lines of the transect are one after the other, do you mean maybe consecutive instead of parallel?

111- Please delete “when amphibians”

120- Please add some information more on the identification you performed based on morphology.

121- Not clear how you used reference 22 here. That reference is about Scincinae. For consistency, you should add analogue references for other reptile groups or modify the whole sentence to simply say that you gave species names based on current taxonomy

123- I suggest using this index only relative to sampling activities (i.e., how common a species was during your sampling) rather than defining a species as common or rare in the community because 5-7 days of sampling are not enough to establish that.

135- Results section. Please, try to consistently use the past tense throughout the result section.

141- “fauna. The species accumulation curves in each site are distinct”. What do you mean exactly? Please, specify

144- Figure 2 legend. Axis names should be in English. Please, write site names instead of site numbers

144- At line 75 you said you visited each site for 7 successive days, but I only see 5 days in Figure 2. Which is the correct number of days?

152- “(50%: 26.1% geckos and 23.8%)”. This is not clear.

156- Please use Paracontias sp. aff. rothschildi instead of Paracontias aff. rothschildi

157- “which with the trap-hole”. Not clear, please rephrase

157- “In other words, the three techniques are complementary for the inventory of reptile.”. This concept is not clear from this sentence. They’re complementary if the methods allowed you to detect different species, as it looks like from Table 2. Please specify this aspect and add more information.

159- 3.2. Distribution and conservation status section.

1-Please add in-text references to Table 2 for species’ IUCN status

2- In this section, you did not fully report the distribution of the surveyed species as expected based on the section title. I suggest that you report here all cases of actual range extension and new distributional records within the known range (see my suggestion at line 198).

161- vulnerable in upper case

165- Pseudoxyrhopus ambreensis is the 14th species, not part of the 13 of the threatened categories. Please, modify the sentence.

165- “In other words, the Complexe Bobaomby represents a potential refuge that can guarantee the protection of these threatened species”. This is a discussion topic

167- “restricted distribution”. How much restricted? If your definition of restricted distribution matches Brown et al. 2014 definition (range less than 1000 km2), then you should use the term “microendemic” here and throughout the text instead of “restricted”. Microendemic is a widely used term in the literature and, by the way, you also included it as a keyword.

Brown, J.L., Cameron, A., Yoder, A.D., Vences, M., 2014. A necessarily complex model to explain the biogeography of the amphibians and reptiles of Madagascar. Nat. Commun. 5, 5046. https://doi.org/10.1038/ncomms6046

168-169- Some of these species are indicated as Endemic to Madagascar and others as Endemic regional in Table 2. Please, correct in the table. Besides, I suggest adding a third distribution category in Table 2 corresponding to these restricted distributed species (again, better microendemic if they follow Brown et al. (2014) definition)

198- Table 2.

1-Have you detected all species with one sampling technique? Did you detect some species with more than one method?

2-Please, add a new column reporting for each species the cases of actual range extension and new distributional records within the known species’ range

3- Please, add a new column indicating for each species the references to manuscripts that reported them in the area of Complexe Bobaomby before your study.

4-“IUCN” instead of “UICN” in the third column.

204- 4.1. Species richness and composition section

1-I suggest using Goodman et al. (2018) to compare reptile species richness between areas of northern Madagascar. The references you used are a bit old and the lists of species reported in Goodman et al. (2018) are surely more updated. Goodman, S.M., Raherilalao, M.J., Wohlhauser, S., 2018. Les Aires Protégées Terrestres de Madagascar: Leur Historie, Description et Biote / The Terrestrial Protected Areas of Madagascar: Their History, Description, and Biota. Association Vahatra, Antananarivo

2-You should acknowledge in the text that you are conscious that these species numbers comparisons with other areas in northern Madagascar are not that informative because different surveys might have used different techniques and efforts.

219- “Ampombofofo site [17]”.

1-In the introduction (line 61) you referenced the previous study on Ampombofofo with Mitchell et al., 2007 [11], while now you're using a different reference. Please, clarify

2-Can you specify if you recorded species that are not reported in the previous study? In general, making a comparison between the two species lists

225- In the introduction, you did not report that Anjiabe was surveyed before your study. Please add that information in the introduction.

230-I would not define this as remarkable. It is normal that you have more preys than predators. Besides, preys relative to whom? Chameleons, lizards and skinks prey on other animals. Please clarify. I suggest you delete this sentence because it does not have relevance.

233- The results are not enough to make this statement. Please, delete this whole reasoning on preys and predators and focus on microhabitat specialisation.

239- Please, specify that Anjiabe and Ampombofofo forests are the least disturbed within the area

247- 4.3. Endemicity and species conservation status section. I suggest that in this section you briefly resume the introduction topic about new PAs that have been designed in rainforest areas, while your study contributes to showing that dry-deciduous forests are rich in reptile diversity and deserve legal protection.

252- Again, do you mean microendemic? Please specify and in case substitute with that term

258- Please, rephrase and argue more on why these three sites are more suitable than the others for conservation purposes.

271- 4.4. Extension of distribution area section. I suggest you incorporate this part into the previous paragraph (lines 247-257) where you already write about distribution and endemism. It is mostly redundant here.

280- Conclusions section. I would like to see a sentence discussing in future perspectives the different results you could obtain in terms of species identification resolution when using molecular data and DNA Barcoding. This is to acknowledge that, although the methodological approach exclusively based on morphology that you used is appropriate for the study aims, a molecular approach is generally more suitable.

Comments on the Quality of English Language

I suggested “Extensive editing of English language required”. Almost all sentences have at least an error related to grammar (e.g., wrong use of verb tenses), typos, missing words and prepositions. There is no need for major rephrasings, but the text requires extensive minor edits in many parts.

Author Response

Thank you for your comments and we tried to adress all of them but we still open for any questions or suggestions.

Reviewer 2 Report

Comments and Suggestions for Authors

The report by Randriamialisoa is a useful addition to the faunal inventory of Madagascar. The authors also discovered a couple of fairly rare species, so these observations are worth reporting. 

I have relatively few issues that should be corrected or improved:

Figure 1 is not labelled as such. The symbols are also difficult to see because mangroves are indicated in black and symbols are black too, so I suggest to use a different color for the symbols.

Line 121: ref 22 is for skinks only; for species with similar species in the same region, authors should explain differences and how they diagnose these species. For example, authors find multiple specimens of Paracontias which they are unable to ID with certainty. Uncertainty is fine, but then they should discuss the differences between known species and how their specimens differ.

Figure 2, labels are in French

Section 3.2. No need to list these species, as they are in Table 2 already. Same for lines 261-265.

Table 2. Please provide raw specimen counts. That's much more informative than generalized abundance categories. E.g. '"Rare" can mean different things. For example, Paracontias minimus is known from very few specimens, so it is interesting that Randriamialisoa et al. discovered several additional specimens. Please summarize the literature about other known specimens. This could be done in section where interesting species are discussed one by one. A comparison with other species is also needed because they have one unidentified Paracontias. Köhler et al. 2009 provide a table with table with traits across the species of Paracontias, so their speciments should be compared to this table. The same should be done for other rare species.

Include photos of such rare species, ideally with photos showing diagnostic traits. 

Ramphotyphlops braminus should be Indotyphlops.

Section 4.1. Authors may indicated species found in the Montagne des Français too, so it's easier to compare the two localities.

Line 230 - blind snakes instead of "Typhlops"

Prey and predatory species is rather arbitrary, as all reptiles are both prey and predatory.

Habitat: Maybe include habitat in Table 2 -- that would summarize all important data in one table.

Section 4.4. List localities in table, supplement, or submit to iNaturalist. For rare or vulnerable species, obscured localities can be used.

Section 6 - Patents. Delete.

Author contributions: delete template stuff, such as "For research articles ..." etc.

Appendix A and B are not specified. Unclear if they exist or what they contain.

Comments on the Quality of English Language

First, the English needs some corrections. I found (mostly small) issues in lines

27 - traps, transects (instead of singular)

32 - classified as ...

33 - three are

34 - three are

45

47-48 - biodiversity face ???

etc.

Some native speaker should read and correct these issues.

Author Response

Thank you very much for your comments. We addressed most of them. Because of the short time we don't have time for a native english speaker but we did some improvements.

Reviewer 3 Report

Comments and Suggestions for Authors

The submitted manuscript is interesting, fits perfectly in the Special Issue, and does contribute for the overall knowledge and need to address conservation efforts toward Madagascar herpetofauna. There are, nevertheless, some issues throughout the manuscript that need to be addressed. From an overall perspective:
- the authors never touch the subject of the ecological importance of reptiles which will help leverage the relevance of the study;
- the sampling strategy is sound but it seems that the collected data was somewhat simplistic from the start. I do understand that the study is a first approach to a previously understudied area but it leaves me with the idea that the study design was unambitious for the planned effort. The authors, for example, could have opted for trying to correlate the presence of the species to its habitats / disturbance / etc. 
- the discussion does not explore in depth the results and there should be a limitation and future perspectives sub-section

Aside from these, I do think the study is relevant and worth publishing, after a moderate revision.

Some issues to look over:
- line 44: please refer to th percentage of the total as previously;
- lines 100-101: please explain why you had four lines, instead of three;
- line 113: ... three 'experienced' herpetologists.
- figure 1: axis names in English
- line 145: the caption needs a more clear explanation
- line 151: ... lizards made up of 21 species... does not make sense
- line 152: 23,8% of what?
- line 153: refer to table 2
- lines 160-165: it is redundant to detail all species when you have a table to summarize everything
- lines 165-167: this sentence belongs to the discussion
- lines 168-170: refer to table 2
- line 208: was the methodology and effort similar between studies?
- lines 212-217: these sentences belong to a limitation and future perspectives sub-section
- lines 232-233: there is a good opportunity to explore the logic behind the natural unbalance between predators and prey
- line 243: why 'harvested'?

Comments on the Quality of English Language

There are some issues throughout the text that need to be solved. I recommend the authors to proofread the manuscript by an native English speaker. 

Author Response

We are thankfull for your comments on our article. We tried to address them and made some improvements. However, we had some difficulties for some of them because of the time.

Round 2

Reviewer 1 Report

Comments and Suggestions for Authors

Dear Authors,

The manuscript has improved compared with the last version, and for this reason I have suggested “Accept after minor revision”. However, for some major comments in my previous review (e.g., a photographic plate of the animals), the authors replied that they will edit the text accordingly, while it is better and common practice to provide a version of the manuscript that already incorporates those changes. In other cases, the authors replied that they changed the text accordingly, but I could not see those changes in the text. Clearly, the authors are not obliged to follow a reviewer’s suggestions. However, they should provide justifications and a manuscript version incorporating suggestions when they decide to follow them. Besides, the English level is still quite poor, as I found numerous grammatical errors. I want to stress that as the current version does not incorporate a photographic plate of the animals, which was the main flaw I highlighted, I consider that the manuscript will not reach publication standard until the plate is added, irrespective of the suggested “minor revision”.

I am reporting the main comments I provided in my last review matching the reasoning I outlined in the above lines, and some additional minor comments:

1-Photographic plates of the animals found during the survey and the visited sites. They are still missing.

2-Line 43. Please, rephrase.

3-Line 46. Please, substitute “expending” with “expanding”.

4-Line 53. Please, substitute “affected” with “protected”.

5-Line 110. Please, substitute “direct observation” with “visual search” for consistency with previous parts.

6- Line 168 – section 3.2. Distribution and conservation status section. In this section, you did not fully report the distribution of the surveyed species as expected based on the section title. I suggest you report all cases of actual range extension and new distributional records within the known range here.

7-Line 176. The authors replied by saying they modified the text according to the suggestion to substitute “restricted distribution” with “microendemic” as a more appropriate term here and throughout the manuscript. However, I did not find these changes.

8-Line 207 – Table 2. Some of these species are indicated as Endemic to Madagascar and others as Endemic regional in Table 2. Please, correct the table. Besides, I suggest adding a third distribution category in Table 2 corresponding to these restricted distributed species (again, better microendemic if they follow Brown et al. (2014) definition).

9-Line 207 – Table 2. The authors said they modified Table 2 according to the suggestion about adding a new column reporting for each species the cases of actual range extension and new distributional records within the known species’ range. However, I did not find these changes.

10-Line 207 – Table 2. Substitute “UICN” with “IUCN”.

11-Line 212 – Discussion session. Can you specify if you recorded species that were not reported in the previous study? In general, making a comparison between the two species lists.

Comments on the Quality of English Language

The English level is still quite poor, with numerous grammatical errors throughout  the text.

Reviewer 3 Report

Comments and Suggestions for Authors

Dear Authors,

Thank you for addressing most of my recommendations. The present version of the manuscript covers almost all of my concerns.

Comments on the Quality of English Language

Final proofreading by an English native is recommended.
